# Should Grain-Based Staple Foods Be Included in Admonitions to “Avoid Processed and Ultra-Processed Food”?

**DOI:** 10.3390/nu17132188

**Published:** 2025-06-30

**Authors:** Julie Miller Jones

**Affiliations:** Professor Emerita, Department of Family, Consumer and Nutritional Sciences, St. Catherine University, St. Paul, MN 55105, USA; jmjones@stkate.edu

**Keywords:** processed/ultra-processed food (UPFs) and health, whole grains and health, NOVA, confounding and processed foods (PFs), NOVA and diet quality and disease risk, processing levels, dietary guidance and whole grains, NOVA and principles of dietary guidance

## Abstract

**Background/Objectives:** The nutritional importance of grain-based foods (GBFs) and whole grains (WGs) is underscored by their central position in dietary guidance worldwide. Many jurisdictions recommend consumers increase WG intake because they are associated with multiple health benefits, with evidence quality rated as moderate to high. High intakes of ultra-processed foods (UPFs), as defined by NOVA that classifies food by level of processing, are associated with numerous negative health outcomes, with evidence less convincing than for WGs. Yet, some dietary guidance recommends consumers to avoid UPFs. This creates two divergent guidelines since NOVA designates most commonly eaten grain-based foods (GBFs) as UPFs. These contradictory guidelines fail to comply with recommended principles of dietary guidance and generate questions about underlying assumptions and definitions that classify WG staples and colas together. **Methods**: Definitions and assumptions for systems ostensibly categorizing foods by level of processing were evaluated for validity by various methods. Special attention was paid to the ability of different classifications to differentiate between WGs, RGs staples, and indulgent GBFs. Findings from meta-analyses associating high intakes of WGs with numerous health benefits were compared with those associating high intakes of UPFs. Menus and modeling studies were assessed for ability to meet recommendations for WGs and the grain food group with customary GBFs while avoiding UPFs. Advice to “avoid UPFs” was tested against principles for effective dietary guidance. **Results**: Definitions and categorizations of foods by levels of processing vary markedly. Assumptions for NOVA and other systems are questionable. While meta-analyses consistently show high intakes of UPFs are associated with adverse health outcomes, high intake of WG foods, nearly all designated as UPFs, are associated with better health outcomes, although evidence quality for the latter is rated stronger. These findings add to the body of evidence suggesting flawed assumptions behind categorizing WG staples in terms of level of processing. **Conclusions**: NOVA deems 90% of WGs as UPFs. Adding statements to dietary guidance to “Avoid UPFs”, while asking consumers to increase WG intakes, confuses. Further, it jeopardizes efforts to increase intake of fiber and WG foods because it excludes top sources of fiber and WG-based breads, rolls, tortillas, or cold cereals in Western diets. NOVA advice to avoid UPFs challenges principles for usable dietary guidance and the construction of culturally appropriate, healthy dietary patterns containing WG staples from all levels of processing.

## 1. Introduction

Grain-based foods (GBFs) and other carbohydrate staples form the basis of dietary guidance worldwide [1]. They earned this position because they are important contributors of calories, plant protein, nutrients such as vitamins and minerals—thiamine, folic acid, iron, magnesium, and zinc, and carbohydrates, including dietary fiber (DF) [2]. Research indicates those ingesting 50% of their calories from these and other carbohydrate staples are associated with having better health and increased longevity than those who ingest higher or lower percentages of calories as carbohydrates [3].

Reviews consistently show that high intakes of whole grains (WGs) are associated with multiple positive health outcomes [4]. Those following dietary patterns with recommended servings of WGs are associated with better nutrient intakes [5]. Thus, dietary guidance advocates consumers increase intakes of WGs by replacing RGs with WG staples to make at least “half of the recommended grain servings” WGs [5]. However, diets often contain more than the recommended number of refined grains (RGs), especially indulgent ones, and too few recommended WG staples. Intakes of WG and DF are far below recommended levels in most countries [6,7].

At the same time, as the body of evidence on WGs was strengthening, Monteiro and his colleagues at the University of Sao Paulo, Brazil, introduced NOVA because they postulated that shifts from traditional foods to nutrient-poor, indulgent foods and beverages—PFs and UPFs—often from multinationals, might be a cause of rising rates of obesity and chronic diseases [8,9].

As a result, they proposed NOVA (not an acronym), a system to categorize foods, ostensibly, by level of processing [8]. NOVA spurred controversy because it deviated from classic definitions and categorizations of food science and technology (FST). Instead of being based on processing complexity, NOVA used criteria such as numbers and presence of detractor ingredients, such as sugar, salt, and additives.

After NOVA's introduction, several alternate categorizations like NOVA were launched, but they often used different definitions, assumptions, and categories, thus adding to confusion of both professionals and consumers. NOVA, the primary focus of this paper, remains the most widely studied and used system and has made its way into frequently quoted dietary advice and even into some dietary guidance even though the evidence associated with UPFs is consistent, but most of it is rated as “weak” or low quality [10]. However, dietary advice to “avoid PFs and UPFs” discounts findings on WGs” positive health impacts because NOVA designates 95% of WG foods as PFs and UPFs. This diametrically opposed advice is troubling. Further, the conflicting findings raise doubts regarding the scientific validity of criteria used to categorize foods according to degree of processing [11]. Further, constructing diets where WGs are primarily from minimally processed foods (MPFs) requires shifts from top contributors of WGs (e.g., breads, rolls, tortillas, and ready-to-eat (RTE) cereals) to GBFs not commonly used in many Western dietary patterns [12].

This paper will present findings that reinforce the importance of GBFs, especially WG staples from the marketplace, as key to building healthy dietary patterns; compare the quality of data from meta-analyses associating high intakes of PFs and UPFs with adverse health outcomes with those showing the opposite for WG foods from the marketplace; and question the validity of criteria for NOVA categorizations, which designate common WG staples as UPFs and place WG staples in the same category with indulgent foods such as sugar-sweetened beverages (SSBs), and the enfranchisement of NOVA advice into dietary guidance. This creates confusion and undermines efforts to increase WG intakes

## 2. Materials and Methods

### 2.1. Food-Based Dietary Guidance for WGs and Definitions

Examples of food-based dietary guidance were selected from the WHO/FAO compilation [1] to showcase the central role of GBFs and WGs in diets around the world (graphics were chosen for their ability to be reproduced, the variety of shapes and foods, and clear depiction of WG foods).

Definitions of PFs and UPFs from FST textbooks, journals, and scientific societies; on-line dictionaries and Wikipedia; government official jurisdiction websites, and public health (PubH) and consumer associations were assessed for similarities and differences. Various classifications and their assumptions were found by searching MedLine (2016–2024) with the terms “processed foods”, “ultra-processed foods, “NOVA” and “food processing.” Documents for the various classifications and those for NOVA from Monteiro and PAHO were scrutinized for details.

NOVA documents were also assessed to capture stated and implied assumptions. These were then measured as to the validity of their scientific bases.

### 2.2. Search Strategies for GBFs and UPFs

Meta-analyses and systematic reviews of prospective cohort studies, and their umbrella reviews, (n = 20) were identified with search terms “whole grains” and “health,” mortality,” “diabetes,” “cancer,” and “cardiovascular disease” using MedLine (2016–2024). These health conditions were chosen because they were the subjects of more studies than other conditions such as dementia. (Sources that analyzed WGs as part of a healthy pattern, such as the Mediterranean diet, were referred to but not directly included.) Medline was also used to identify 16 meta-analyses and systematic reviews of prospective cohort studies associating intake of UPFs and “health”, “mortality,” “diabetes,” “cancer,” and “cardiovascular disease”. Several umbrella reviews of UPFs and adverse health outcomes were consulted for analyses of evidence quality. Further, a few papers cited in one meta-analysis were chosen to show examples of confounding for which adjustments were likely inadequate. Papers on UPFs and diabetes and mortality that had subgroup analyses of WGs as UPFs were also used to provide evidence for the likely mis-categorization of WG staple foods. Other references were sought to address specific points.

### 2.3. Strategies for Testing Assumptions

Twenty GBF labels from recommended and non-recommended GBFs (6 PFs and 14 UPFs, half of which contained WGs) were analyzed to test whether the number and types of ingredients are valid criteria for labeling a food “UPF”.

The appendix NIH randomized controlled trial (RCT) of NOVA [13], was used to assess numbers and types of GBFs used to meet the WG and grain food group recommendation. Also, a modeling study was identified to provide data on the impact of diet and WG intake when UPFs are removed. Further, advice to “avoid PFs/UPFs” was evaluated against principles for effective dietary guidance (as outlined by Rowe et al.) [10].

## 3. Results and Discussion

### 3.1. Grain-Based Foods in Dietary Guidance

GBFs have prime importance in dietary guidance for most countries. These foods, along with other carbohydrate staples, provide the bulk of calories and needed nutrients [1,2], including B vitamins, especially thiamin and folic acid, iron and other minerals, such as magnesium and zinc, and DF. Grains” ability to complement amino acid profiles of other plant foods such as legumes makes them key to plant-forward diets [14]. Guidance recommends that about 50% of the calories come from GBFs and other carbohydrate staples. Such diets are associated with better health and greater longevity than those containing higher or lower percentages of carbohydrate calories [3].

Figure 1 depicts food-based dietary guidance for a few countries. Despite cultural differences in both foods and formats, these infographics show that GBFs have prominent roles in diets worldwide [1]. GBFs often personify a culture, e.g., corn tortillas in Central America, pasta in Italy, baguettes in France, rice in Asia, grainy breads in northern Europe, and flat breads in the Middle East and South Asia.

While all dietary guidance recommends GBFs, not all have recommendations regarding WGs. A 2019 analysis found WGs recommendations throughout North America and in 70% of European and Near Eastern countries, but in less than half of the remaining countries [15,16,17]. Never-the-less, efforts to increase WGs intake are on-going because strong evidence associates intakes of 50 gWG/d (~three 30 g servings of WGs food) with better nutrition, reduced disease risks, and improved health [7,18,19,20].

### 3.2. Grain-Based Foods Are Associated with Better Health and Lower Disease Risks

Meta-analyses of epidemiological studies involving large cohorts find those ingesting the most WGs are associated with reduced risks of many conditions, including all-cause and cause-specific mortality [21,22,23,24,25,26,27,28]; type 2 diabetes mellitus [26,29], cardiovascular and coronary heart disease, stroke and heart failure [22,23,24,25,26,27,28,30,31,32,33], and many cancers, with the strongest evidence for colorectal cancer [25,26,27,33,34,35,36,37].

Recent umbrella reviews determined that the strongest evidence exists for associations between WG intake and the prevention of type 2 diabetes and colorectal cancer [38,39]. WGs and grain fiber intakes were strongly associated with lower colorectal cancer risks, but total DF was not [40]. High WG intakes correlate strongly with high DF intakes, and both are also associated with improved diet quality, reduced inflammation, and disease risks [41].

An analysis of U.S. National Health and Nutrition Examination Study (NHANES) intake data shows that GBFs, both WGs and RGs, provide all the cereal fiber and over half of the total DF in the diet [42]. Two of three umbrella reviews found WG intakes strongly associated with lower risks of overall mortality, cardiovascular disease, and pancreatic and gastric cancers, with potential benefits occurring with intakes of at least 45 g WGs/day [43,44].

Studies show WGs are important contributors to healthy dietary patterns such as USDA MyPlate or the Mediterranean diet, partly because they supply cereal fiber and phytochemicals [45] and are associated with improvements in many aspects of health [46]. The observed positive health benefits associated with WG intakes are based on marketplace foods, most of which are deemed PFs and UPFs in NOVA.

### 3.3. NOVA and New Definitions of Processed Foods

#### 3.3.1. Definitions of Processed Foods—FST and Official Jurisdictions

PFs were classically defined by FST professionals, but that changed in 2010 when Monteiro and colleagues launched a PF definition and classification with a PubH perspective. However, PubH definitions bear little congruence with those in FST textbooks and journals [47,48], or those issued by professional associations such as the International Union of Food Science and Technology [49,50]. FST defines PFs as those whose natural state has been altered by one or more physical or chemical processes to transform raw ingredients into edible foodstuffs. FST professionals determine the processing level by the number and complexity of operations, so foods that are minimally processed have undergone few operations. They hold that foods are processed for the following reasons:To extend the time food remains microbially and biochemically wholesome and safe to eat.To maintain or enhance nutrient content or availability of foods.To remove or inactivate any harmful constituents.To provide a variety of marketable foods that can enhance taste and convenience.To increase sustainability of the food supply [50].

WHO/FAO notes food processing makes foods edible, improves quality, and extends the food supply [51]. Like those from authoritative bodies such as FAO/WHO, definitions from other regulatory and authoritative bodies align with FST. The U.K. National Health Service (NHS) and Food Safety Authority Australia New Zealand (FSANZ) note that most foods are processed in some way, with changes ranging from minimal to substantial [52,53]. The European Food Safety Authority (EFSA) stipulates that processing applies to all stages of production, processing, and distribution of food [54]. The U.S. Department of Agriculture (USDA) added that processing may improve eating quality and involve the addition of other ingredients such as preservatives, flavors, nutrients, and approved additives. The USDA stated that processing may reduce, increase, or leave unaffected the nutritional characteristics of raw agricultural commodities [55]. Codified definitions of PFs used by regulatory agencies in most jurisdictions align with FST [51,52,53,54,55,56,57,58] and therefore do not define UPFs.

The Pan American Health Organization (PAHO), a subsidiary of FAO/WHO, is the exception because its definitions align with those promoted by Monteiro and other PubH groups [8,9]. PAHO's definition makes no reference to the complexity of processing, only to the number and types of ingredients. Foods with sugar, salt, other additives, or four or fewer ingredients are PFs. Those with more than four ingredients are UPFs [59,60].

#### 3.3.2. Definitions of Minimally Processed Foods

MPFs are classically defined by FST as involving few operations of minimal complexity. Very few jurisdictions officially define MPFs. The USDA defines MPFs as foods that have been slightly altered but without significant changes in nutritional contribution [54]. In contrast, MPFs are defined in most PubH categorizations. PAHO adopted the NOVA definition that states MPFs are the edible parts of plant or animal foods without other ingredients and treated with processes that retain the food's innate properties. The French system (Siga) by Fardet defines MPFs as natural foods that have had inedible parts removed or have been dried, crushed, filtered, heated, cooled, or packaged, or have undergone nonalcoholic fermentation [61]. However, Siga's definition does include some impacts of processing, such as degree of food matrix destruction as well as ingredients added (Section 3.4).

#### 3.3.3. Definitions of Ultra-Processed Foods—Official Jurisdictions

The term “UPFs” is not defined by FST professionals, but they describe “highly processed” foods as those undergoing many processes, some of which may be highly technical or complex. Further, the level of processing is unaffected by the addition of ingredients/additives or place or scale of processing. Further, more processing does not inherently lower nutritional or health aspects.

UPFs are not officially defined in most jurisdictions, although they are mentioned in some dietary guidance. For example, Canada's Food Guide suggests limiting consumption of “highly processed” foods and beverages and preparing more food at home. Health Canada's examples of highly processed foods are mostly non-recommended, indulgent, salty, sugary foods. In terms of GBFs, breads were not on the list, but breakfast cereals were [61]. The U.K. Food Standards Agency (FSA), EFSA, and FSANZ have no official definitions but note they are assessing health impacts of highly processed foods [62,63,64,65]. FSA suggests that consumers consider, rather than processing per se, the nutritional value of various foods and their contribution to overall diet and health and follow existing recommendations [64].

#### 3.3.4. Definitions of PFs and UPFs—Dictionary and Encyclopedia

Encyclopedia and on-line dictionary definitions of PFs vary. Wikipedia defines PFs as food resulting from food processing and notes that processing transforms agricultural products, such as raw wheat kernels, into familiar foods such as bread [66]. The entry notes ancient processes such as threshing, winnowing, and milling and modern processes such as factory dough production. On-line dictionaries such as Collins and Merriam Webster both define PFs as those subjected to processes during manufacture, often to preserve them [67,68]. Both definitions have a portion that aligns with FST but also a portion that aligns with PubH. Collins [67] states that UPFs are prepared from multiple ingredients using complex industrial methods, some having little or no nutritional value. Merriam Webster defines UPFs as highly processed ingredients including artificial additives (such as coloring, flavoring, and preservatives) and typically having high levels of fat, sugar, or salt [68].

While Wikipedia defined PFs under “Food processing”, it also has an entry for UPFs, with the introductory caveat “there is no simple definition of UPFs” [6]. It defines UPFs as foods resulting from relatively involved methods of production, which may be industrial creations of natural foods or synthesized components. Thus, parts of the definition align with PubH and part with FST definitions. The Oxford English Dictionary does not define MPFs or PFs but added a definition of UPFs in 2024 [68], stating that UPFs have “a high degree of industrial processing” and “large quantities of additives” such as 'salt, sugar, fat, preservatives, or artificial colours and flavourings”. Thus, it aligns definitions from other on-line dictionaries and a PubH perspective. It includes descriptive material stating “UPFs have sensory qualities of unprocessed or minimally processed foods, while disguising undesirable qualities of the final product” or food robbed of nutrients.

#### 3.3.5. Definitions of Processed and Ultra-Processed Foods—Health Promotion Organizations

Definitions from most health promotion organizations align with PubH and focus on detractor ingredients, not processing complexity. For example, the U.K. NHS defines them as foods “undergoing more processing than other foods” and possibly including ingredients not usually found in home settings, such as preservatives, sweeteners, and emulsifiers. Among the GBFs on the UPF list recommended “to avoid,” cakes and biscuits (cookies) appear, but bread does not.

The American Institute for Cancer Research (AICR) states that food processing is any deliberate change in a food, through simple or complex processes. UPFs often contain ingredients not found in home kitchens that extend shelf life or make foods ready-to-eat or heat [68]. Detractor nutrients (salt, sugar, fat, or additives) raise the level of processing. UPFs are defined as prepared with industrial equipment and added ingredients, some of which are not found in home kitchens [69,70].

University of North Carolina (UNC) researchers defined “processed food” as everything from washed vegetables to candy and sodas. The initial part of the definition aligns with FST, but its categorizations align closely with NOVA, although unlike in NOVA, RG foods are considered more processed than WGs [71].

The Organic Consumers Organization or results from consumer survey responses suggest consumers define PFs or UPFs as “widely distributed foods that aren’t natural or whole that contain additives (e.g., artificial or modified ingredients or synthetic flavorings) and provide convenience” [72,73]. Consumers believe MPFs are close to their original state and nutrient content, and have few detractor ingredients, but they struggle to differentiate between PFs and UPFs. Consumers state tenets such as “the longer a food's ingredient list, the more likely it is a UPF” [74].

### 3.4. Categorization of Processed Foods by Various Systems

NOVA was the first categorization system defining level of processing not using FST principles. It is the most used and studied system, but after its launch, other systems followed (Table 1). They are included and discussed briefly because they demonstrate the lack of agreement around definitions, categories, and assumptions in the treatment of RG and WG foods.

#### 3.4.1. Categorizations Compared for Assumptions and Treatment of GBFs

Various categorizations use different definitions and assumptions. FST and regulatory bodies do not assume that the place of preparation, number or type of ingredients or additives, processing procedures even when the native matrix is disrupted, or packaging automatically change either the level of processing or nutrient and health contribution of foods [48]. Any of these may positively or negatively impact nutrient availability or retention, food safety and eating quality, shelf life, and food cost.

The International Agency for Research on Cancer (IARC) categorization has 3 main categories [79]. Only foods that are eaten as kernels or their crushed forms are “Minimally Processed” (e.g., WG kernels, flours, meals, and grits). If these or a component isolated from the whole kernel such as bran are used, the food is “Highly Processed.” Refined and enriched flours, rices, pastas (not couscous, but couscous is a pasta), and isolated brans or germs are not treated as ingredients as in NOVA but are categorized with products made from them such as breads, crackers/biscuits, breakfast cereals, commercial dough, and popcorn and are “Highly processed.” This categorization fails to differentiate between indulgent or recommended GBFs or WG and RG breads and cereals, as they are all deemed “Highly Processed”.

The International Food Information Council’s (IFIC) categories most closely align with FST because they were generated by a joint task force of food scientists and nutritionists. This categorization assumes that neither number nor types of ingredients/additives and place or scale of production inherently impact health outcomes. Terms such as “highly” or “ultra” processed were not used; instead, categories reflect reasons for processing or end use. Convenience and packaging are not viewed as inherently negative. Analyses of NHANES databases using the IFIC categorization demonstrate that dietary patterns, not individual foods or level of processing, determine nutrient intakes and diet quality [81,82,83]. Modeling showed that healthy diets could be constructed with foods from all IFIC levels of processing. GBFs in IFIC are found in all categories. Both WG and RG breads and RTE-cereals are classed as “packaged foods ready to eat” [81].

The International Food Policy Research Institute (IFPRI) was developed for mid- and low-income countries to improve diets [84]. It is also an easily understood system because highly processed foods are mostly indulgent foods, so recommended foods are not categorized with non-recommended foods. WGs and RGs may be in the same category. It is similar to the Mexican National Institute of Public Health’s system, which has two categories—”Unprocessed” and “Processed”. The latter is further divided into two subcategories: “Processed Traditional” and “Processed Modern”. This system was based on assumptions that shifts from traditional to modern food patterns contributed to poor diets and rising obesity rates [85,86]. WGs and RGs are not clearly differentiated in this system. Unprocessed GBFs would include whole kernels and their ground and nixtamalized forms. These would be related to MPFs in several other systems. Tortillas, breads, or other foods made from these ingredients would be “Processed”. However, the place of preparation may determine if foods are “Processed Traditional” or “Processed Modern”. If corn tortillas are home prepared, they would be “Processed Traditional”, but corn tortillas prepared commercially and all things prepared with wheat flour including wheat tortillas are “Processed Modern,” so WG and RG breads, pastas, and RTE-cereals would be “Processed Modern.” The system does not differentiate WGs and RGs.

Siga [79,80,81,82,83,84,85,86], with its three categories and multiple subcategories, is complicated. Its basic assumption is that loss of intact raw matrix reduces healthfulness. It is true that more nutrients may be lost with finer particles, but nutrient availability may increase with more processing, such as finer grinding or making products such as flakes or biscuits, which results in more vitamin and mineral bioavailability than from whole kernels [87,88,89]. Protein foods, including GBFs, may be more or less digestible with changes in the matrix [90].

Siga considers the presence of additives and detractor ingredients to raise the level of processing. Siga differs from other systems in that the amount of detractor ingredients, not merely their presence, places foods into different subcategories [91]. Siga’s authors specifically warn that “industrialized foods” with healthy halos such as gluten-free or micronutrient-enriched” are ultra-processed [84]. In Siga, WGs and RGs may be differentiated because of grain particle size and matrix. However, RG and WG breads or cereals might be classed as having the same level of processing, depending on the amounts of salt, sugar, and additives.

The University of North Carolina (UNC) system by Poti et al. has four main categories with multiple subcategories [78]. Its basic assumption is that nutrient density, especially for highly fortified foods, masks negative food components that impact health. The “Unprocessed” category includes single foods—WG rices and crushed whole grains, such as oatmeal or popcorn prepared with no sugar, salt, or fat. Only a few GBFs, such as classic French baguettes or RTE-cereals meet criteria for “Moderately Processed”. Most GBFs breads, tortillas, RTE-cereals, crackers, granolas, and bars are “Highly Processed” because all require other ingredients for structure. While the UNC categorization might help consumers distinguish between a few WG and RG foods, it fails to do so for most GBFs because the presence of added ingredients make them “Highly Processed” regardless of their WG content.

#### 3.4.2. The NOVA Categories Outlined

Details of the NOVA categorization are found in Table 2. NOVA was based on the premise that dietary shifts from traditional to marketplace foods, especially those heavily marketed by multinationals, diminished diet quality and increased obesity in Latin America [8,9]. NOVA categories are based on number and types of ingredients. In NOVA, single foods with nothing added are MPFs. Processed Culinary Ingredients (PCIs) are defined as pastas and flours and pantry items in a common kitchen. PFs are those with four or fewer ingredients, usually an MPF with PCIs, and/or contain sugar, salt, or other additives, and UPFs have five or more ingredients. In NOVA, 90% of commonly eaten GBFs are UPFs, 5% PFs, and 5% MPFs and PCIs combined [11].

#### 3.4.3. The NOVA and GBFs

NOVA classifies most GBFs with markedly different assumptions than FST. For example, FST would deem nixtamalized corn, RTE vacuum-packed rice, or other grains with nothing added or instantized kernels (e.g., instant rice) as PFs because of their processes, but NOVA deems them MPFs. However, the addition of sugar, salt, flavorings, or inclusion of four or more ingredients in NOVA makes foods UPFs even if the level of processing is unchanged. Thus, MPFs, PFs, and UPFs might have identical processes applied and only differ by the presence of one or more ingredients. FST classifies most fresh or dry pastas as PFs, but NOVA classifies them as PCIs. Unless they are in mixes with other ingredients, then they are UPFs. The number of ingredients is the main determinant of whether foods are PFs or UPFs (Table 2).

Table 3 presents a convenience sample of 20 GBFs that are a mix of PFs and UPFs, WG, and RG, and recommended and non-recommended foods with ingredient statements and some nutrition information. It also shows the number and type of ingredients and fortificants, fiber, sugar, salt, calories from the nutrition panel, WG status, NOVA categorization, and whether the food is recommended by dietary guidance.

Of the 20 GBFs, 15 would be staples and five non-recommended, indulgent foods. There are six PFs (four ingredients)—one WG bread store brand (SB), one RG French bread; two WG cereals “a“ and “b”, and two RG cereals containing sugar, salt, and malt. (Both RG cereals contain malt, which is a natural ingredient that bakers have in their pantry. It is unclear how if it would be categorized by NOVA as a normal kitchen ingredient.) Of the PFs, the WG bread and WG cereal would be recommended in dietary guidance, but the RG cereals and French bread would be recommended sometimes as staple foods to make “half your grains whole”. (Note: Several PFs meet the rules for ingredient number, but WG cereal “b” contains BHT and flavor, so it might be deemed UPF. Both RG cereals (corn flakes and crisp rice) have four ingredients making them PFs, but they have eight fortificants, possibly causing them to be labeled as UPFs).

Thirteen of the 20 contain WG, but only 12 would be deemed a WG food because the quinoa candy does not have WG as the first ingredient, and it is an indulgent food. Of these 12 (nine UPFs and three PFs), all but two would be recommended without reservation. The two RTE WG cereals (UPFs) would raise concerns due to sugars, maybe sometimes (S) placing them in the ‘S’ category. Two PFs and nine UPFs, including the chocolate quinoa candy, contain more than 3 g of fiber per 60 g (the equivalent of serving two slices of WG bread) and 10% of the DF daily value, so a good source. Thus, the designation of UPF and PF would be unlikely to help consumers choose recommended GBFs and might impair the intake of WGs.

### 3.5. NOVA and Assumptions for Categorization

NOVA’s categorization fails to distinguish between WGs and RGs and recommended and non-recommended GBFs. Such findings raise questions regarding assumptions for NOVA categorization. Table 4 contains NOVA’s assumptions, both stated and inferred derived from explanatory material in supporting documents containing instructions for food placement and quotes such as the following [8,9,59,92]. UPFs are “attractive, hyper-palatable, cheap, ready-to-consume food products…characteristically energy-dense, fatty, sugary or salty and generally obesogenic….Industrial formulations of processed food substances…that contain little or no whole food and typically include flavourings, colourings, emulsifiers, and other cosmetic additives” [9,59,93,94].

The first assumption (Table 4) that UPFs contain no real food is wrong. The 20 foods in Table 3 all contain WG or RG ingredients, and 17 of 20 have a grain, flour or flour mixture, or seed as the first ingredient, so these do contain “real food”. One has dark chocolate as the first ingredient. Even those that list sugar as the first ingredient or contain ingredients not found in home kitchens contain real food ingredients.

Approximately one-third contained color, flavors, emulsifiers, and/or preservatives not found in home kitchens. Many WG foods also contain nuts or bran etc., and despite their contribution of MPFs, such as seeds and partial kernels of grain, innate fiber, and nutrients, they would be UPFs. Three contain the natural preservative tocopherol (an isomer of vitamin E), four contain a chemical preservative (BHA or BHT) or propionate, and four contain colors and or emulsifiers. Perhaps a more reliable indicator of foods to avoid would be having sugar as the first ingredient rather than erroneously stating that UPFs have no real food. If consumers wish to avoid additives, these are on the ingredient statement, so using them as criteria for avoidance seems unnecessary.

The second assumption that foods with higher levels of processing are less nutritious (Table 4) is also not demonstrated in Table 3. The UPF multigrain bread has more fiber, nutrients, WGs, nuts, and seeds than the RG baguette or either of the RG cereals that are PFs. Further, NOVA places most RG and WG foods in the same category, even though the removal of bran and germ layers makes RGs more processed. For most breads, the mixing and baking processes are the same whether PF or UPF. If anything, the WG breads may require more gentle kneading, and some additives such as emulsifiers may mean less kneading or mixing is required.

The third assumption (Table 4) that number of ingredients makes a foods UPF has little scientific or logical basis [95]. The WG multi-seed bread (with 35 ingredients) and the WG crispbread or WG bread-a (each 17 ingredients) are UPFs but would be recommended choices, with all being more nutritious choices than the four-ingredient PF examples of RG French bread, corn flakes, or crisp rice cereal in Table 3, or even single-ingredient PCIs/MPFs such as hominy grits or farina. While the snack cake with 27 ingredients or the chocolate quinoa candy with seven ingredients are less nutritious choices than corn flakes with four ingredients, none of these would come close to matching the nutrition found in the WG multi-grain cereal with 12 ingredients. Ingredient number does not prove to be a valid criterion for level of processing or a touchstone for healthfulness.

The fourth assumption (Table 4), namely that detractor nutrients—sugar, salt, or saturated fat—in any amount make a food UPF, lacks basis. Dietary guidance is about total daily intakes. For example, daily sugar intakes should be <10% of energy. It makes little sense to apply these criteria to a single food when recommendations are for total diets. Hess and colleagues [96] modeled diets with UPFs that meet the sugar limitation. Designating foods with any level of detractor nutrients as either PFs or UPFs reduces incentives for manufacturers to reformulate foods to lower the amounts of these constituents. For example, no- sugar or low-salt versions of WG crackers would both be UPFs, despite differing sugar levels.

The fifth assumption (Table 4), i.e., that foods containing additives should be avoided, disregards scrutiny by various authoritative bodies. Some foods including GBFs may be safer and more sustainable with additives. Emulsifiers and preservatives such as those in some cereals or breads (Table 3) help maintain softer crumb or better texture, extend shelf life, lower cost, and improve sustainability, but they reduce oxidation, fat rancidity, and inhibit molds, all of which can be harmful to the body.

The sixth assumption (Table 4), that place and scale of preparation influence healthfulness of food, is debatable. Nutritional quality and food safety may be superior in commercial operations because processes can be optimized and monitored to protect labile nutrients and ensure food safety. Such protocols are tightly regulated in commercial facilities; thus, the majority of foodborne outbreaks are related to home- or restaurant-prepared foods [97].

The seventh assumption (Table 4), that foods produced for special diets, whether required (e.g., for medical or life-stage needs (e.g., lactose or gluten-free or blood sugar controlling foods, infant formula or supplements for the infirm or elderly)) or preferred, such as vegan or plant-forward, are to be avoided because they are UPFs, is not helpful to those requiring or wanting these foods. Their availability can improve the nutrition and lives of those with various conditions [98]. In terms of infant formula, breast feeding is always preferred, but formula is necessary in situations where it is not possible. Foods for the very young, including those that are UPFs, have been shown to have important roles in supporting nutrition [99].

The eighth assumption (Table 4), that UPFs have been engineered to be “hyperpalatable”, does not reflect data on WG foods [100]. Sensory data collected for WG foods consistently demonstrate that color, texture, and flavor are barriers for consumers choosing to replace RGs with WGs and increase WG intakes [101]. These foods are rarely accused of being “hyperpalatable” or the cause of overconsumption.

The ninth assumption (Table 4), that convenience, availability, affordability, and sustainability are reasons to avoid a food, shows a lack of understanding of consumer needs and situations. Many sectors of the population benefit from foods that are readily available, affordable, and convenient, and all benefit from less food waste.

The tenth assumption (Table 4) is that packaging is another indicator of a higher level of processing. While plastics and packaging do add to the waste stream, but so do stale, rancid, oxidized, contaminated, or moldy foods. Packaging reduces staling and contamination of all kinds and improves flavor and nutrient retention. Packaging is found on foods with any level of processing. For example, MPFs in NOVA such as fresh greens, vacuum-packed cooked beets, or cooked rice may be found in pouches, cans, and Tetra Paks.

The eleventh assumption (Table 4), that processing level overrides a food’s nutritional content or place in dietary guidance, defies logic [102,103]. The nutritional contribution of PFs in Table 3, or MPFs such as farina or hominy, is lower than that of most of the WG breads and cereals in Table 3. Thus, dietary recommendations that advocate such foods over WG recommended UPF staple foods misguide consumers.

The twelfth assumption (Table 4), that MPFs and homemade foods are better for building healthy diet patterns, may or may not be valid. While MPFs can be prepared in creative ways with minimal sugar and salt, they may also be prepared from fatty meats and with many ingredients that leave them with nutritional profiles that are far less salutary than those of many UPFs [104,105]. While salt and sugar amounts can be controlled in home preparation and diets may be improved with home cooking, there can be marked variation in food choice and the use of discretionary ingredients [106,107,108].

### 3.6. Difficulties Incurred When Meeting WG Requirement with Only MPFs

Meeting recommendations for 5–6 servings/day of GBFs with only MPFs proved difficult, as demonstrated by analyzing the week’s menus for RCT by Hall and colleagues at NIH. Subjects ate either diets with 85% UPFs or 85% MPFs [14]. Though this pilot study was criticized for its short duration and significant differences in factors known to impact food intake, (e.g., food volume, chewing, eating duration, DF types), little comment has been made about not meeting food group recommendations. While diets were matched for nutrients, menus in the MPF arm failed to meet MyPLATE recommendations for the bread and cereal group on 5 of 7 days per week. Specifically, no GBFs were included on 1 day or in MPF snacks on any day; a GBF was served on 1 day in one meal (RG basmati rice); GBFs were served on 3 days in two meals (bulgur, basmati rice; WG quinoa, barley; WG quinoa, penne); and GBFs were served on 2 days in each of the three meals (WG oatmeal, farro, couscous; WG oatmeal, barley, spaghetti). Further, GBFs may not have allowed meeting the WG recommendations. However, it is not clear because menu descriptions and depictions [14] Appendix did not state whether foods were WGs. During the MPF phase of the experiment, subjects on most days would be unlikely meet recommendations for cereal fiber, WGs, of total number of grain servings.

The MPF menus contained neither any handheld GBFs nor top sources of WGs in Western diets—breads/rolls, tortillas, RTE cereals [13], and wheat were not served on 3 of 7 days despite wheat being the most commonly consumed grain in many countries [109]. When wheat foods were served, they were pastas (including couscous), bulgur (common with those from the Mediterranean), or the wheat parent farro, an “ancient grain” with limited availability.

Modeling exercises using the Australian food intake database confirmed that omission of PFs and UPFs resulted in decreased intakes of WGs, DF, cereal fiber, folate, iodine, iron, and B vitamins [110]. While the authors of this study and others noted that it would be possible to meet the WG requirement with MPFs, they stated that such shifts would necessitate dramatic, and probably unlikely, changes in cultural food choices [102,111,112]. Thus, both the intervention menus and modeling studies illustrate the difficulty of constructing diets using commonly eaten GBFs when PFs and UPFs are avoided.

### 3.7. NOVA, Whole Grains, and Health Outcomes

Systematic reviews and meta-analyses of epidemiological studies using NOVA [11,103,113] and their umbrella reviews [114,115] consistently associate high intakes of UPFs with over 50 adverse outcomes, such as inferior diet quality [116] and increased risks of chronic disorders, such as type 2 diabetes (T2D) [117,118], dyslipidemia, and cardiometabolic risk factors [119], various cancers [120,121], depression and mental health disorders [122], dementia [123], Crohn’s and inflammatory bowel disease, [124], non-alcoholic fatty liver disease [125], chronic kidney disease [126], hypertension [127], weight gain, obesity [128], and all-cause mortality [129]. Metabolic syndrome may be associated with UPF intakes, but data show significance only in case–control but not in prospective cohort studies of adults [119].

Results touting ill effects of UPFs have been widely publicized in the popular, nutrition, and public health press. Some have conflated associations to causations and stated that the evidence is overwhelming [130]. While such findings raise alarm about diets high in UPFs, neither the low strength of the evidence nor potential error sources and confounders are reported. Evidence quality in several meta-analyses was rated as “weak” or received GRADE ratings as “low” [11,108,118]. For example, in Lane et al. [11], “using the GRADE framework, 22 pooled analyses were rated as low quality, with 19 rated as very low quality and four rated as moderate quality”. Further, while there is damning evidence for high intakes of UPFs, few report that moderate intakes may not raise risks and that a number of foods rated as UPFs, including breakfast cereals and breads, lowered risks [21].

#### NOVA Health Outcomes Differ for Total UPFs and Foods Deemed UPFs

Studies also show that individual UPFs analyzed separately change the findings markedly. High total UPF intakes in the EPIC cohort and three US health professional cohorts were associated with a 46% increase in risk of T2D [117]. However, high intakes of UPF breads, biscuits, and breakfast cereals analyzed separately were associated with up to a 35% decrease in T2D risk. Similar, but less dramatic findings occurred in the US cohort [118]. High intakes of total UPFs raised T2D risk by 25% of more, but RTE-cereals were associated with 22% decreased risk, while high intake of WG breads was associated with only a 4% drop [118]. The differences observed between the two studies may be due to types and amounts of WG breads eaten in Europe and the US. Similar patterns were seen for all-cause mortality where breakfast cereals reduced mortality risks by 15% [129]. Such findings fit with the large body of literature showing such GBFs, especially WGs, are associated with lower risks of T2D and overall mortality [21,22,23,24,25,26,27,28,29,30,31,45].

Subgroup analyses of various UPFs provide another piece of evidence suggesting the erroneous categorization of recommended foods such as WG staple foods with non-recommended SSBs, processed meats, and savory snacks. Separate analyses show that high intakes of these foods were associated with increased risk of T2D, of from 25% to 277%. Thus, these findings align with the large body of literature showing that SSBs or processed meats increase T2D risk [39], and with those showing WGs are associated with lower risk (reviewed in Section 3.2).

### 3.8. NOVA and Potential Sources of Error

#### 3.8.1. Sources of Error in NOVA—Mis-Categorization

The many non-congruent definitions and categorizations of PFs and UPFs lead to mis-categorization (Section 3.3, Table 1 and Table 2). Studies document that food and nutrition professionals have difficulty placing food into categories [131,132]. Error chances increase when food intake instruments not designed for use with NOVA are used to categorize foods, as information may be insufficient for accurate placement.

Observed associations with various health outcomes and the degree of risk may be impacted by mis-categorization. This was shown in a comparative study by Vitale et al. that showed that while high intakes of UPFs, regardless of system used to categorize the foods, were associated with higher risks, the relative risk of most diseases changed as various systems categorizing processed foods were used [113].

#### 3.8.2. Sources of Error in NOVA—Confounding

Confounding, which is inadequately recognized or adjusted for, may be a source of error [133]. In studies of UPFs, there are often numerous co-variates and significant confounders that are significantly different between groups ingesting high intakes of MPFs and UPFs or that vary markedly by socioeconomic status (SES), education, and income [134]. Properly adjusting for these and their potential interactions is daunting. The possibility of inadequately adjusted or unrecognized confounders is high [118,135]. For example, Mendonça et al. acknowledged this difficulty in their paper associating high intakes of UPFs with increased risk of hypertension [136], noting that the findings were heavily confounded by multiple dietary factors and differences in SES and education level. The latter co-vary with a cluster of factors such as limited access to recommended foods and other resources (Section 3.8.2) [137]. Studies consistently show that those with low SES and less education may eat few WGs, fruits, and vegetables and choose more RGs and non-recommended foods [96,138].

### 3.9. NOVA, GBFs, and Meeting Criteria for Dietary Guidance That Changes Behavior

Dietary principles that promote positive behavior change and encourage the building of healthy diets were outlined by a group of expert nutritionists and food scientists [10]. The first principle is that recommendations should be clearly understood by health professionals and consumers and give explicit guidance regarding food choices. The numerous operative definitions and categorizations of PFs and UPFs (Section 3.3; Table 1 and Table 2) demonstrate the lack of clarity regarding foods to choose. Further, studies demonstrate that nutritionists or researchers applying NOVA found food categorization confusing and difficult, and inconsistent ratings were prevalent [96,131,132]. While most consumers and professionals correctly categorize SSBs and candies as UPFs, professionals showed a high degree of inter-rater variability for most other foods [139]. Despite inconsistent categorization of foods, except SSBs, into NOVA, half of UK adults reported that their food choices were determined by whether they “believe” a food is a UPF [140]. Many were unaware that 95% of GBFs, including WGs, are PFs or UPFs [131,132,141]. Consumers in one survey believed that “UPF” should refer to foods subjected to numerous industrial processes or containing additives or artificial ingredients [142].

Correct categorization influences study outcomes, as demonstrated by analyzing PrediMed intake data by four different categorization systems [143]. While high intakes of UPFs resulted in poor diets regardless of system, associations with disease risks differed, showing that placement into categories matters.

Dietary guidance is effective if recommendations are actionable, affordable, and practical for all segments of the population. Several studies substantiate that diets comprising mostly MPFs cost more [13,111,138,144,145,146]. USDA studies show dramatic differences in costs (2023 pricing) of MPF and UPF diets, of USD 34.87 vs. USD 13.53 per day per person [147]. Asking consumers to avoid most fortified WG bread/cereals (<20 g sugar/100 g), which rank as top contributors of nutrient-dense, low-cost foods, and replace them with uncommon, more expensive and less available MPFs such as farro and quinoa abrogate these principles [148,149]. Meal patterns, as demonstrated in the USDA’s Thrifty Meal Plans that include foods from all levels of processing, should be encouraged especially for food-insecure populations [96,111].

Beyond cost, there are many barriers to adoption of diets containing mostly MPFs. Home-prepared GBFs, such as WG breads or tortillas, require skill, equipment, and time. Time is a huge constraint for all but may have the greatest impact on those with lower SES and working women [150]. In 1900, when there were no labor-saving devices or UPFs, few women worked outside the home, and USDA estimates showed that the average American woman spent 44 h/week (farm activities such as gathering eggs were not included in this calculation) on food preparation and clean-up [151,152]. Today 70% of U.S. women (20–64 years of age) work outside the home, so diets using only MPFs challenge most. Situations limiting time for food preparation are many and include holding two jobs with long hours or commutes, care responsibilities for children, elderly, or the infirm, time needed for additional education or training, and physical or mental limitations, to name just a few. Other barriers include limited market access and lack of skill, adequate cooking facilities, or spaces to properly store food. Full endorsement of MPFs without caveats about the nutritional contribution of added ingredients or cooking methods may do little to improve diet quality [153,154]. Further, unless dietary advice informs and motivates consumers about better overall choices such as increased consumption of fruits, vegetables, and whole grains and balancing calorie intake with need, diets may not improve.

### 3.10. Treatment of Ultra-Processed Food and WGs Differs in Dietary Guidelines

Half the world’s dietary guidelines mention processing, but many only in the context of processed meats [155]. Belgium, Brazil, Canada, Ecuador, Israel, Maldives, Peru, and Uruguay give specific advice about UPFs or “highly processed” foods, but that advice varies [156]. For example, the 2023 Uruguayan Guidelines state: “Base your diet on natural foods and avoid the regular consumption of ultra-processed products with excessive contents of fat, sugar and salt”. Thus, Uruguay modifies the statement with “avoid regular” consumption of UPFs and defines them as foods high in fat, sugar, and salt. The 2020 Canadian Guidelines state: “Choose whole foods and minimally processed foods often. Limit highly processed foods”. They also used the word “limit”, not “avoid”, and the examples are sugary drinks, fast foods, and deli meats. WG bread is depicted prominently in Canada’s Food Guide (Figure 2). In contrast, the 2021 Brazilian Guidelines recommend consumers “Avoid ultra-processed products and ready-to-eat foods”. They cite bread (wheat flour, yeast, and water) as an example of a UPF to avoid. Thus, guidance about UPFs and GBFs by authoritative bodies conflicts.

## 4. Limitations

This analysis has several limitations. It is not a systematic review. MedLine was used as the bibliographic source. Thus, the analysis may not have adequately covered all definitions or categorizations of UPFs or found all studies on WGs and UPFs.

Since most categorizations used ingredients as a critical factor to increase the level of processing, the review tried to show that this assumption was not well supported by scientific principles. The convenience sample of GBFs (Table 3) to demonstrate the numbers and types of ingredients in WG and RG foods included PF and UPF, WG and RG, and recommended and non-recommended categories. It is neither random nor exhaustive. This may affect the results but also clearly shows the number of ingredients is not a supportable assumption for determining level of processing. A 60 g amount was used for comparison because it can be used for breads (two 30 g slices) as used in US labeling. Most GBFs have a 30 g serving size, and much of the world uses a 100 g serving size, so some might have used one of these. The use of 60 g might have unfairly raised the amounts of sugar and salt for foods, which are normally 30 servings, and doubled their DF.

The determination of the WG status of the MPF diets in the Hall paper [14] was based on the appendix pictures and descriptions of the food, which did not state if the pastas or rices were WGs, so it was assumed they were not. Since NOVA does not differentiate WGs and RGs, they might be either. Further, it was assumed that the recommended number of servings of WGs and GBFs would not be ingested on days when grains were in a single meal, and it is questionable if they would eat three servings in each of two meals. This could be in error, as the feeding was ad libitum.

The implied assumptions for NOVA may have been improperly deduced from documents about NOVA.

The meta-analyses for both UPFs and WGs are epidemiological studies and have the potential for many confounders and misclassifications, which would affect the findings. The reporting of data quality or evidence GRADE was not found in all the meta-analyses, so the reliance on a few papers might result in error.

## 5. Conclusions—UPFs, NOVA, Dietary Guidance, GBFs, WGs, and Health

The introduction of NOVA, a system that ostensibly classifies foods by level of processing, has spurred controversy since its inception. Its basic assumptions, definitions, and categories were based on numbers and types of ingredients, not level of processing. Numerous meta-analyses associate high intakes of UPFs with poor diet quality and risks of over 50 adverse health outcomes. However, these findings conflict with meta-analyses from the same cohorts showing WGs from the marketplace, of which over 95% are deemed PFs and UPFs in NOVA, with improved diet quality and numerous better health outcomes.

Many public health professionals and some dietary guidelines recommend the avoidance of UPFs. This creates contradictory dietary advice because staple GBFs are a key component of food-based guidelines, with many actively encouraging increased consumption of WG foods.

The naming of staple GBFs as UPFs by NOVA raises questions regarding NOVA’s basic assumptions. When GBFs were separated from other UPFs such as sodas and processed meats in meta-analyses, most were associated with lower disease risks. These findings align with the large body of evidence associating WG breads and RTE WG cereals with lower disease risks. Further, assessments of modeling and daily menus constructed without UPFs showed the near impossibility of meeting GBF and WG requirement with customary GBFs. Additionally, when assumptions such as number of ingredients or presence of additives were tested in a sample of GBFs, the exercise demonstrated the inability of NOVA to differentiate between recommended and non-recommended GBFs.

Numerous definitions and categorizations around UPFs cause confusion for both professionals and consumers. This lack of clarity violates the first principle of dietary guidance. Guidance also needs to be not only understood but workable—affordable, practical, and culturally appropriate—to evoke effective dietary change.

WGs and staple GBFs from the marketplace are proven components of healthy dietary patterns and take center stage in dietary guidance worldwide because they are convenient, low-cost contributors of carbohydrates, DF, plant protein, and micronutrients. Intakes of both WGs and DF in nearly every country are below recommendations. Dietary advice, such as that in NOVA, which recommends avoidance of top dietary contributors (breads, cereals, and tortillas) of DF, WGs, and key nutrients, could impair diet quality and health. Stigma resulting from labeling culturally important staples as UPFs and advising their avoidance would impede efforts to improve dietary patterns and increase consumption of these healthy, staple WG foods.

## Figures and Tables

**Figure 1 nutrients-17-02188-f001:**
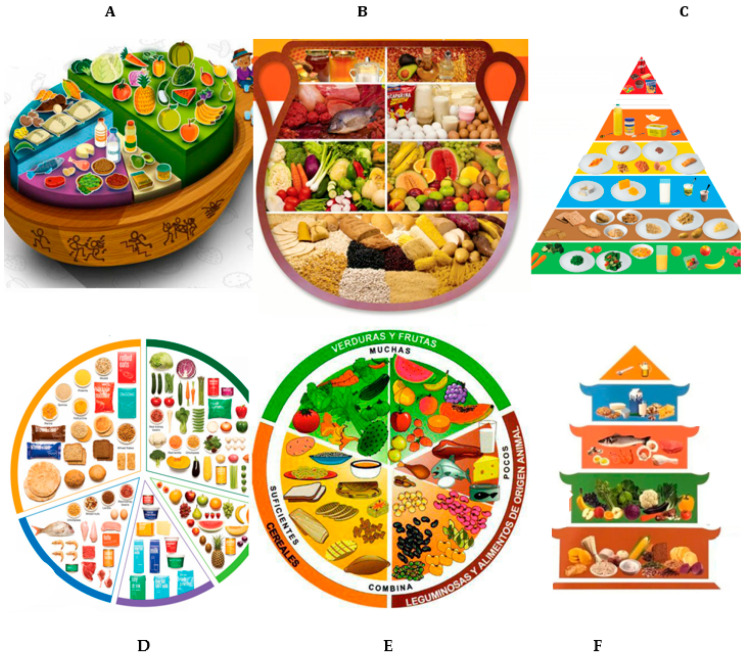
Examples of the prominent role of grains and grain-based foods in dietary guidance as depicted by these samples from the FAO/WHO compilation [1]. (**A**) Food-based dietary guidelines of Ecuador (A Wooden Spoon). Spanish: *Guías alimentarias basadas en alimentos del Ecuador*); (**B**) Dietary guidelines for Guatemala. Recommendations for healthy eating. The Family Pot. (Spanish: *Guías alimentarias paraGuatemala. Recomendaciones para una alimentación saludable*); (**C**) Irish Healthy Food for Life—the Healthy Eating Guidelines and Food Pyramid; (**D**) Australian Guide to Healthy Eating; (**E**) The Mexican Plate of Good Health (*El Plato del Buen Comer*); (**F**) Balanced Diet Pagoda for China.

**Figure 2 nutrients-17-02188-f002:**
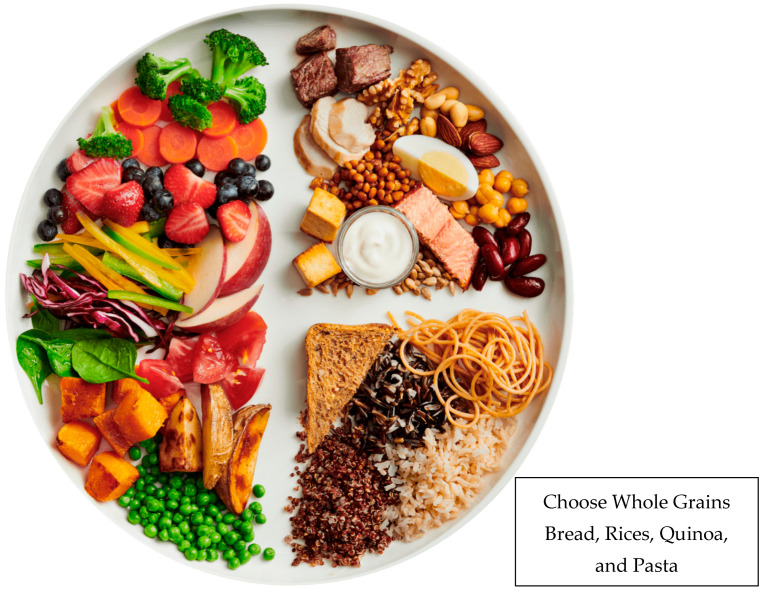
Canada’s 2025 Food Guide—Health Canada [1].

**Table 1 nutrients-17-02188-t001:** Systems of Categorizing Food by Degree of Processing.

Agency/Organization	Treatment of Grain Foods	Notes
International Agency for Research on Cancer (IARC) [70].	Nearly all grain-based foods (GBFs) are highly processed. Only cooked grains and broken cooked grains are deemed minimally processed foods (MPFs).	IARC classifies more foods as ultra-processed foods (UPFs) than other systems.
International Food Information Council (IFIC) [75].	Grains are in every category. Whole grains (WGs) may be in the same category as refined grains (RGs).	IFIC categorizes food by end-use and is most aligned with food science.
The International Food Policy Research Institute (IFPRI) [76].	Most WG breads and cereals are processed foods (PFs). WGs and RGs may be in the same category as RGs.	IFPRI’s treatment is designed to be understood by consumers because highly processed foods are mostly indulgent foods.
The Mexican National Institute of Public Health [77].	Tortillas prepared at home are “traditional.” Bread, rolls, or tortillas prepared with refined flour at home are “processed traditional.” If purchased, they are “processed modern.”	The Mexican system is relatively simple and differentiates by place of preparation and whether refined flour is used.
NOVA [8,9].	WGs are classed with RGs. 90% of commonly eaten GBFs including WGs are UPFs; 5% PFs; and 5% PCIs and MPFs combined.	See Section 3.4.2
University of North Carolina (UNC) [78].	GBFs are classed similarly to NOVA, except WGs are not included with RGs.	The UNC system developed by Poti using public health criteria.
Siga—Institut National de la Recherche Agronomique (INRA) France [79,80].	Only whole kernel grains are rated as minimally processed. Each step causes matrix breakdown and increases processing levels. Sugar, salt, and fat and number of ingredients raise processing level. WGs and RGs may be classed differently because RGs have greater matrix loss. However, they may not be if they contain detractor ingredients or additives.	Siga posits that changes in the food matrix are unhealthy. It has three main categories, each with subcategories that depend on degree of matrix disruption and level of detractor ingredients.

**Table 2 nutrients-17-02188-t002:** NOVA Categorization of Foods by Level of Processing and Effect on Whole and Refined Grain Foods [92].

NOVA Category	Grain Foods per Category
**GROUP 1: Unprocessed/Minimally processed** foods are treated by cleaning, removal of inedible parts, fractioning, grinding, drying, fermenting, pasteurizing, cooling, freezing, or other processes. No added oils, fats, sugar, salt or ingredients are allowed.	Whole kernels of wheat, oats, corn, rices, or other grains and their crushed and broken counterparts—grits, flakes, and flours from kernels. These may be fortified with iron, folic acid, or other nutrients lost during processing or instantized.
**Group 2 Processed culinary ingredients (PCIs)**—These are common ingredients found in a kitchen pantry, such as sugar, fat, flours, leavenings, spices, and flavorings.	Dried or fresh pasta including couscous, polenta, grits, flakes, and flours.
**Group 3 Processed foods (PFs)**—These are MPFs or PCIs that contain salt, sugar, oil, or other substances to preserve or improve palatability. They are recognizable versions of the original foods. Most have four ingredients.	Freshly made, (unpackaged) breads made of flour, yeast, water, and salt. Breads, cereals, frozen items with four ingredients. (It is unclear if one of the four ingredients is a preservative or emulsifier if the food is PF or UPF.)
**Group 4: Ultra-processed foods (UPFs)**—These are industrial formulations made mostly from substances extracted, modified, or synthesized constituents (oils, fats, hydrogenated fats, sugars, starches, proteins, flavors and enhancers, colors, and other additives that make the product hyper-palatable). MPFs comprise only a small proportion or are absent from these products.	Packaged bread, especially if in plastic bags, and baked products made with ≥5 ingredients, such as hydrogenated vegetable fat, sugar, yeast, whey, emulsifiers, and other additives; breakfast cereals and bars; canned, packaged, dehydrated noodles; sugary, savory or salty packaged snacks; biscuits/cookies; pastries, cakes, bakery and savory side mixes; pre-prepared rice, grain, pizza, and pasta dishes; dinner rolls, tortillas, and other ethnic flat breads, hamburger and hot dog buns.

**Table 3 nutrients-17-02188-t003:** Selected Whole Grain (WG) and Refined Grain (RG) Processed Foods (PFs), Ultra-processed Foods (UPFs), Ingredient Number, Statements, and Selected Nutrient Information.

Grain Food with Number of Ingredients (I) and Fortificants (F)			Ingredient Statement	Sugar (g)	Dietary Fiber (g)	Sodium (mg)	PF (P) UPF (U) # Recommend Y, N, S
Food	I	F		Per 60 g serving ^†^			
WG Bread- NB (National brand)	17	-	Whole Wheat Flour, Water, Wheat Gluten, Sugar, Yeast, Vegetable Oil (Soybean), Salt, Preservatives (Calcium propionate, Sorbic acid), Datem, Natural Flavors, Monoglycerides, Monocalcium phosphate, Soy Lecithin, Citric Acid, Grain vinegar, Sesame Seeds.	2	4	240	U Y
WG Bread- SB (Store brand)	4	-	Whole Wheat Flour, Wheat Flour, Water, Salt.	0	5	270	P Y
WG multigrain bread	35	-	Organic Whole Wheat (Organic Whole Wheat Flour, Organic Cracked Whole Wheat), Water, Organic Cane Sugar, Organic 21 Whole Grains And Seeds Mix (Organic Whole Flax Seeds, Organic Sunflower Seeds, Organic Ground Whole Flax Seeds, Organic Brown Sesame Seeds, Organic Triticale, Organic Pumpkin Seeds, Organic Rolled Barley, Organic Rolled Oats, Organic Rolled Rye, Organic Black Sesame Seeds, Organic Blue Cornmeal, Organic Millet, Organic Rolled Spelt, Organic Brown Rice Flour, Organic Amaranth Flour, Organic Yellow Cornmeal, Organic KAMUT^®^ Khorasan Wheat, Organic Quinoa, Organic Buckwheat Flour, Organic Sorghum Flour, Organic Poppy Seeds), Organic Wheat Gluten, Organic Oat Fibre, contains 2% or less of each of the following: Organic Molasses, Sea Salt, Yeast, Organic Vinegar, Organic Cultured Wheat Flour, Enzymes, Organic Acerola Cherry Powder.	12	12	510	U Y
WG Multi-seed, crisp bread *	11	0	Sunflower Seeds, Sesame Seeds, Wholemeal Rye Flour, Oat Bran, Oatmeal, Flaxseed, Spelt, Wheat Bean, Wheat Bran, Water, Sea Salt, Salt Oregano, Thyme	0	8	300	U Y
RG French bread	4	0	Organic Unbleached Flour, Water, Sea Salt, Organic Barley Malt.	0	1	280	S
RG Enriched bread SB	6	6	Unbleached Enriched Flour (Wheat Flour, Malted Barley Flour, Niacin, Reduced Iron, Thiamine Mononitrate, Riboflavin, Folic Acid), Water. Contains 2% or less of: Sour culture, Salt, Yeast, Semolina.	0	<1	360	U S
WG Cereal-a	4	0	Whole Grain Wheat, Cane Sugar, Cinnamon, Natural Flavor.	9	7	0	P Y
WG Cereal-b	3	0	Whole Grain Wheat, Wheat Bran, BHT.	0	8	0	P Y
WG Oat cereal	5	0	Organic Whole Grain Oat Flour, Organic Wheat Starch, Organic Cane Sugar, Sea Salt, Calcium carbonate, Mixed Tocopherols.	4	6	300	U Y
WG Cereal-c	6	8	Whole Grain Wheat (96.1%), Sugar, Invert Sugar Syrup, Barley Malt Extract, Salt, Molasses. Iron, Vitamin B3, B5, B9, B6, B2.	8	8	400	U Y
WG Cereal-d	9	8	Wheat (50.8%) Whole, (37.4%) Rice, Oats Integral (7.1%) Sugar, Partially inverted Sugar Cane Syrup, “Extract Malt Barley, Vitamins/Minerals: Calcium carbonate, Niacin, Iron, Acid pantothenic, Vitamin B6, Riboflavin, Folic acid; Salt, Glucose Syrup, Antioxidant Tocopherol-rich extract. (4% Sugars/Serving).	6	10	52	U Y
WG Multigrain cereal	12	0	Khorasan Wheat Flour, Wheat Bran, Whole Wheat Meal, Cane Sugar, Whole Oat Flour, Spelt Flour, Barley Flour, Whole Millet, Barley Malt Extract, Quinoa, Sea Salt, Honey Organic.	6	7	200	U Y
WG Cereal-e *	17	8	Corn Flour Blend (Whole Grain Yellow Corn Flour, Degermed Yellow Corn Flour), Sugar, Wheat Flour, Whole Grain Oat Flour, Modified Food Starch, Contains 2% or less of Vegetable oil (hydrogenated coconut, soybean and/or cottonseed), Oat Fiber, Maltodextrin, Salt, Soluble Corn Fiber, Natural Flavor, Red 40, Yellow 5, Blue 1, Yellow 6, BHT for freshness. Vita-mins and Minerals: Vitamin C (Ascorbic Acid), Reduced Iron, Niacinamide, Vitamin B6 (Pyridoxine Hydrochloride), Vitamin B2 (Riboflavin), Vitamin B1, (Thiamin Hydrochloride), Folic Acid, Vitamin D3, Vitamin B12.	17	5	300	U S
WG Cereal -f * with Apple	24	8	Corn Flour Blend (Whole Grain Yellow Corn Flour, Degerminated Yellow Corn Flour), Sugar, Wheat Flour, Whole Grain Oat Flour, Modified Food Starch, Contains 2% or Less of Vegetable Oil (Hydrogenated Coconut, Soybean and/or Cottonseed), Oat Fiber, Salt, Soluble Corn Fiber, Degerminated Yellow Corn Flour, Dried Apples, Apple Juice Concentrate, Cornstarch, Cinnamon, Natural Flavor, Modified Corn Starch, Yellow 6, Wheat Starch, Baking Soda, Yellow 5, Red 40, Blue 1, BHT for Freshness. Vitamins/Minerals: Reduced Iron, Niacinamide, Vitamin B6 (Pyridoxine hydrochloride), Vitamin B2 (Riboflavin), Vitamin B1 (Thiamin hydrochloride), Folic Acid, Vitamin D3, Vitamin B12.	22	4	430	U S
RG Corn Flakes	4	8	Milled Corn, Sugar, Malt Flavor, contains 2% or less of salt. Vitamins and Minerals: Iron (ferric phosphate), niacinamide, vitamin B6 (pyridoxine hydrochloride), vitamin B2 (riboflavin), vitamin B1 (thiamin hydrochloride), folic acid, vitamin D3, vitamin B12.	6	1	400	P S
RG Crisp Rice	4	8	Rice, Sugar, Salt, Malt Flavor, and fortified with vitamins and minerals iron, niacinamide, vitamin B6, vitamin B2, vitamin B1, folic acid, vitamin D3, and vitamin B12.	4	<1	240	P S
RG Sweet roll	27	6	Enriched Flour (wheat Flour, Malted Barley Flour, Niacin, Reduced Iron, Thiamine Mononitrate, Riboflavin, Folic Acid), Water, Sugar, Liquid Sugar (sugar, Water), Butter (pasteurized Cream, Salt), Eggs, Contains less than 2% of: Potato Flour, Yeast, Whey, Nonfat Milk, Soy Flour, Salt. Degerminated Yellow Corn Flour, Wheat Gluten, Sodium Sterol Lactylate, Datem, Monocalcium Phosphate, Wheat Flour, Calcium Sulfate, Sodium Silicoaluminate, Ascorbic Acid, Ammonium Sulfate, Wheat Starch, Sorbitan Monostearate, Enzymes, Micro Crystalline Cellulose, Calcium Silicate.	11	1	170	U N
RG Snack Cake	27	6	Sugar, Corn Syrup, Enriched Bleached Flour (Wheat Flour, Niacin, Reduced Iron, Thiamin Mononitrate [Vitamin B1], Riboflavin [Vitamin B2], Folic Acid), Water, Palm and Palm Kernel Oil, Palm and Soybean Oils with TBHQ and Citric Acid to Protect Flavor, Dextrose, Soybean Oil, Contains 2% or Less of Each of the Following: Dried Egg Whites, Corn Starch, Whey (Milk), Leavening (Baking Soda, Sodium Aluminum Phosphate), Cocoa, Salt, Mono- and Diglycerides, Sorbitan Monostearate, Soy Lecithin, Sorbic Acid (to Preserve Freshness), Titanium Dioxide, Polysorbate 60, Natural and Artificial Flavors, Polysorbate 80, Annatto Extract, Turmeric Extract, Soy Flour.	37	0	180	U N
RG Crispy Rice and Cocoa Nib Snack Bars	9	3	Sugar, Corn Syrup, Enriched Bleached Flour (Wheat Flour, Niacin, Reduced Iron, Thiamin Mononitrate [Vitamin B1]. Cane Sugar, Cocoa Butter, Ground Oats, Unsweetened Chocolate, Puffed Rice (rice, cane sugar, salt, sunflower lecithin [emulsifier]), Cocoa Nibs, (rice syrup, sunflower lecithin (emulsifier), Sea Salt, Natural Vanilla Flavor.	18	<1	180	U N
Dark Chocolate Cherry Quinoa Candy *	7	0	Dark Chocolate (Cacao Beans, Pure Cane Sugar, Cocoa Butter, Vanilla Beans), Quinoa, Cherries, Sugar, Sunflower Oil.	10	5	0	U N

^†^ 55–60 g/serving is 2 slices of bread. Smaller servings would be used for some foods in the chart such as the multi-seed crisp bread serving is 1 piece (24 g) or the chocolate quinoa candy. # Recommended by dietary guidance Y = Yes, N = No, or S = Sometimes. * Larger than USDA serving. Cereals ‘a’–‘f’ were so designated in order to differentiate them but avoid using brand names.

**Table 4 nutrients-17-02188-t004:** Stated and Implied Assumptions for NOVA Categorizations [9,59,93,94].

1. Highly processed foods are food-like substances that contain little “real” food.
2. Foods with more processing are less healthful.
3. Number and types of ingredients determine level of processing. Foods with fewer ingredients, regardless of their nutrient contribution or function, are less processed.
4. Any level of sugar, salt, and saturated fat added to food increases the processing level.
5. Additives, regardless of function, raise processing level.
6. The place and scale of preparation determine level of processing.
7. Foods that are hyperpalatable cause their overconsumption.
8. Processing that lowers costs or increases convenience and sustainability is not beneficial. 9. Packaging inherently raises level of processing. 10. Processing level by supersedes a food/food groups nutritional contribution or role in the diet. 11. MPFs and homemade food regardless of their method of preparation, ingredients, or nutritional contribution are always acceptable

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
