# Peer review of "Should Grain-Based Staple Foods Be Included in Admonitions to “Avoid Processed and Ultra-Processed Food”?"

_nutrients, 2025, doi:10.3390/nu17132188_

Round 1
Reviewer 1 Report
Comments and Suggestions for Authors
My recommendations are the following:
Abstract – I recommend mentioning that it is a narrative review article.
I recommend that the keywords be deleted - Definitions and assumption, it is not relevant and is too general.
I recommend reviewing the method of inserting bibliographic indexes, because it creates confusion. You have an index mentioned in parentheses and then another number, probably the same index.
Lines 42-43 I recommend mentioning the bibliographical sources that support the idea.
Introduction – I recommend that at the end of this section the purpose of this study and the main objectives or hypotheses be mentioned. In this sense, I recommend moving the aspects mentioned in lines 67-79, because they are objectives, not part of the Methods section.
In the Methods section – I recommend mentioning the typology of the study in the section called Study design, which I recommend adding. I also recommend mentioning the method of selecting meta-analyses and bibliographic reference sources, in the Procedures section.
Fig 1 is not clear, the writing cannot be understood, I recommend enlarging it.
Table 2 I recommend mentioning in the first line what the information related to each column is.
Table 3 I recommend rewriting respecting the front and size.
I also recommend that the title of table 3 not be inserted in the table, but above it, idem table 4.
I recommend that under table 4 I mention descriptively what the acronyms represent.
The last column of table 4 - Indulgent (I) or Recommended Staple ®, I recommend that R be written as a capital letter distinguished by correcting it. The table does not fit on the page, I recommend revising. The last three lines, in the second column, probably this was not consistent when copying, because the information is shaded, I recommend correcting it.
Line 391 is a confusion in inserting bibliographic indexes. I recommend again the author to be careful with the way of writing.
I recommend mentioning the limitations of this study.
In conclusion, this article has involved a lot of work of substantiation, my recommendations mainly concern aspects of editing and organization of information.
Author Response
|
Thank you for your belief in this paper. I do feel it is needed when NOVA is gaining so much traction. .
Comment 1: Abstract – I recommend mentioning that it is a narrative review article Response 1: Agree: I did this Response 2: Comment I have made deletions of the general terms and changed keywords to be specific to the manuscript. Comment 3: I recommend reviewing the method of inserting bibliographic indexes, because it creates confusion. You have an index mentioned in parentheses and then another number, probably the same index. Comment :5 Introduction – I recommend that at the end of this section the purpose of this study and the main objectives or hypotheses be mentioned. In this sense, I recommend moving the aspects mentioned in lines 67-79, because they are objectives, no longer part of the Methods section Response 5: This is a good idea. I have made the moves and placed the objectives at the end of the introduction. Comment 6: In the Methods section – I recommend mentioning the typology of the study in the section called Study design, which I recommend adding. I also recommend mentioning the method of selecting meta-analyses and bibliographic reference sources, in the Procedures section. Comment 7: Fig 1 is not clear, the writing cannot be understood, I recommend enlarging it. Comment 8Table 2 I recommend mentioning in the first line what the information related to each column is. Response 8: Table 2 is now Table 1. It has been streamlined to focus on the impact of grains and not so much detail on the other categorizations. Response 9: Table 3 which is currently Table 4 has the Title is part of the table. The instructions said you could use Font 8 for much information. Response 10: Table 4 is now Table 3. It now fits the page. Now the columns match and headings changed. The column uses Yes (Y), No (N) and Maybe (Y/N) and these are labeled not I and R. Response 11: This has been changed. See lines 593-602 where the various diseases are tied to the health outcome with the reference near.
I think thou |
Reviewer 2 Report
Comments and Suggestions for Authors
I have read the article entitled:
Should Grain-Based Staple Foods Be Included in Admoni-2 tions to “Avoid Processed and Ultra-processed Food?” » that is a narrative review The subject is very interesting but the paper is not suitable for publication in its present form.
The objectives of the research are too broad, there is a lack of focus. There were understood from the general text but they should be stated more clearly in both the abstract and the introduction.
Τhe structure of the text and the writing are very confusing even for a narrative review. Tables presentations are also unclear.
Author Response
The subject is very interesting; but the paper is not suitable for publication in its present form.
Your comments were taken to heart and I thank you for them. It needed revision and reworking which I have worked to do
The objectives of the research are too broad, there is a lack of focus. They were understood from the general text but they should be stated more clearly in both the abstract and the introduction.
I realize I am trying to cover several areas that seem disparate but impact WGs very adversely. One reason this occurs is the assumptions and categorizations for PFs and UPFs are are not looked at critically.
- I tried to give a topline summary of the importance of GBFs and WGs in diets around the world, and in so doing show their nutritional, dietary and cultural contribution through dietary guidance in many countries. Their importance and cultural role of certain foods are brough into question ‘to avoid PFs and UPFs’ when systems categorize 95% of GBFs including WGs (breads, cereals and tortillas) are deemed PFs and UPFs. Such designations hinder efforts to change consumers from consumption of too many indulgent RGs foods and replace them with WG staples and increase the consumption of WGs. I think that this needs to be addressed and it was done in this paper.
- In addition, the multiple definitions and categorizations of PFs and UPFs, ostensibly by level of processing, but really by detractor ingredients – not only confuse, but impede efforts to increase WGs intake.
- The blanket inclusion of nearly all GBFs as UPFs fails to consider the rich literature associating WGs and health, especially when the weak and low-quality evidence of UPFs and health is compared with the stronger quality of evidence for WGs.
- Few have scrutinized assumptions and logic behind NOVA’s classifications, such as numbers of ingredients or presence of additives, that result in colas, infant formula, and WG bread being placed in the same category. Thus, the exercise in this paper does that.
- Admonitions about UPFs are creeping into dietary guidance despite their non-concordance with principles of dietary guidance. This is also addressed. Guidelines in the same document to increase WG intakes and reduce PFs and UPFs are contradictory and therefore confusing.
I feel that these must be addressed in the same paper and therefore I did create a very difficult problem for myself.
Τhe structure of the text and the writing are very confusing even for a narrative review. Tables presentations are also unclear.
I, with the help of reviewers’ comments, have attempted to re-organize the paper to try to reduce confusion. I have taken out Table 1 and talked about it in the narrative. I have tried to make the tables clearer.
Reviewer 3 Report
Comments and Suggestions for Authors
This narrative review discusses an actual and important terminological question from a broad perspective and in an expert manner.
Remarks
There are two simultaneous systems of citations, one in square brackets, another in superscript. Please eliminate one (obviously the superscripts).
“NOVA’ or “Nova”, please uniformize
Figure 1. The resolution of the Figure is not sufficient, but it may be a question of the pdf quality. Anyhow, please check at the proofreading stage
Line 113 “DF”, please explain acronyms on the first use (fortunately, a list of abbreviations is provided)
Table 4: Please explain “sv”
Line 388 and next: metabolic syndrome could be mentioned
Reviewer 4 Report
Comments and Suggestions for Authors
Evaluation of manuscript nutrients-3633472
This is an interesting review on whole food grains and health outcomes. Below, I present my feedback to help the author improve the manuscript.
Text Aesthetics: The authors should prioritize improving the manuscript’s visual presentation. Figure 1 is unintelligible due to poor image resolution. If the figure cannot be integrated into the text with acceptable clarity, I recommend moving it to an appendix.
Tables: None of the tables adhere to formatting standards. Please revise them before resubmission. Text within the tables is overly lengthy. Consider summarizing content or converting non-essential tables into text to save space. Evaluate whether all tables are truly necessary.
Abstract: The results section in the Abstract lacks depth and does not reflect the complexity of the manuscript. Expand this section to better represent the study’s findings.
Clarity of Objective: The study’s objective needs to be stated more clearly.
Methods: Include the number of texts analyzed under each inclusion criterion to enhance transparency.
References: The current formatting of references does not comply with Nutrients journal guidelines. Please revise accordingly.
Final Remarks
This review holds significant potential but requires thorough revisions to meet publication standards. Addressing the above points will strengthen the manuscript’s clarity, presentation, and scientific rigor.
Author Response
|
Response to Reviewer 4 Comments
|
|
||
|
1. Summary: |
|
|
|
|
Evaluation of manuscript nutrients-3633472. This is an interesting review on whole food grains and health outcomes. Below, I present my feedback to help the author improve the manuscript. Response 1: Thank you very much for taking the time to review this manuscript. Please find the detailed responses below. I did not use track changes because I reorganized so much of the paper almost everything bled red.
|
|
||
|
2. Questions for General |
|
||
|
Agree: I tried very hard to reorganize, make the work clearer, and less misleading. I thank both you and the other reviewers for suggestions to improve the organization and strengthen the paper. |
|||
|
|
|
|
|
|
3. Point-by-point response to Comments and Suggestions for Authors |
|
||
|
Comment 1: Text Aesthetics: The authors should prioritize improving the manuscript’s visual presentation. Figure 1 is unintelligible due to poor image resolution. If the figure cannot be integrated into the text with acceptable clarity, I recommend moving it to an appendix. Response 1: Agree: I originally had trouble with the template. I think I really did work hard to improve text aesthetics in line with journal format, make the paper more readable and less confusing. Figure 1 was fixed by my son, who regularly does graphics for his job. He needed to use figures where the original image (not the thumbnail) was available to get enough pixels. In order to add identifiers and keep them intelligible, we used fewer examples, but they tell a consistent story and represent Asia, South America, Central America, and Europe. I moved the North American one (Health Canada) to be Figure 2 in the last section, where I refer to the bread in the image between lines 730 -732 . All should meet the standard for needed resolution, so I will leave it in the body of the paper and not move it to the Appendix. |
|
||
|
Comment 2: Tables: None of the tables adhere to formatting standards. Please revise them before resubmission. Text within the tables is overly lengthy. Consider summarizing content or converting non-essential tables into text to save space. Evaluate whether all tables are truly necessary. Response 2: I Agree. I changed all the tables to concur with formatting standards. I deleted the original Table 1 about definitions completely and put the material into the text in Sections 3.3.1 to 3.3.4 as suggested by another reviewer. I shortened significantly what was Table 2 (now Table 1). Since the focus of the paper is on NOVA the most used system, so I gave fewer details about alternate systems. The purpose of showing categorizations and definitions is to show inconsistencies among the systems among researchers. This means confusion for all. The table about assumptions and barriers to using MPFs could be removed but I think tables help to break up the long article and emphasize how many assumptions behind NOVA are flawed. Comment 3: Abstract: The results section in the Abstract lacks depth and does not reflect the complexity of the manuscript. Expand this section to better represent the study’s findings. Response 3: Abstract has been reworked and includes the finding Comment 4: Clarity of Objective: The study’s objective needs to be stated more clearly. Response 4: I have added the objectives at line 91 to the end of the introduction Comment 5: Methods: Include the number of texts analyzed under each inclusion criterion to enhance transparency. Response 5: Inclusion criteria have been added to the methods section Comment 6: References: The current formatting of references does not comply with Nutrients journal guidelines. Please revise accordingly. Response 6: Reference format is corrected. Comment: Final Remarks Response to final comments: Thank you for your kind words and time you took to evaluate my paper. It is my hope that I used the comments from you and the other reviewers to strengthen the paper and the objectives are clearer
|
|
||
For review article
|
Response to Reviewer X Comments
|
||
|
1. Summary |
|
|
|
Thank you very much for taking the time to review this manuscript. Please find the detailed responses below and the corresponding revisions/corrections highlighted/in track changes in the re-submitted files. [This is only a recommended summary. Please feel free to adjust it. We do suggest maintaining a neutral tone and thanking the reviewers for their contribution although the comments may be negative or off-target. If you disagree with the reviewer's comments please include any concerns you may have in the letter to the Academic Editor.]
|
||
|
2. Questions for General Evaluation |
Reviewer’s Evaluation |
Response and Revisions |
|
Is the work a significant contribution to the field? |
[Please give your response if necessary. Or you can also give your corresponding response in the point-by-point response letter. The same as below] |
|
|
Is the work well organized and comprehensively described? |
|
|
|
Is the work scientifically sound and not misleading? |
|
|
|
Are there appropriate and adequate references to related and previous work? |
|
|
|
Is the English used correct and readable? |
|
|
|
3. Point-by-point response to Comments and Suggestions for Authors |
|
|
|
Comments 1: [Paste the full reviewer comment here.]
|
||
|
Response 1: [Type your response here and mark your revisions in red] Thank you for pointing this out. I/We agree with this comment. Therefore, I/we have….[Explain what change you have made. Mention exactly where in the revised manuscript this change can be found – page number, paragraph, and line.] “[updated text in the manuscript if necessary]” |
||
|
Comments 2: [Paste the full reviewer comment here.] |
||
|
Response 2: Agree. I/We have, accordingly, done/revised/changed/modified…..to emphasize this point. Discuss the changes made, providing the necessary explanation/clarification. Mention exactly where in the revised manuscript this change can be found – page number, paragraph, and line.] “[updated text in the manuscript if necessary]” |
||
|
4. Response to Comments on the Quality of English Language |
||
|
Point 1: |
||
|
Response 1: (in red) |
||
|
5. Additional clarifications |
||
|
Please see the selected file. Thanks so much for your time and effort in making this paper better. I do appreciate it. |
||
Round 2
Reviewer 2 Report
Comments and Suggestions for Authors
The revisions made by the authors in response to the reviewers’ comments have significantly improved the clarity, rigor, and overall quality of the work. The manuscript is suitable for publication in its present form.
Author Response
Comment: The revisions made by the authors in response to the reviewers’ comments have significantly improved the clarity, rigor, and overall quality of the work. The manuscript is suitable for publication in its present form.
Response to Reviewer:
Thank you for doing the second evaluation of manuscript nutrients-3633472. I know, especially because the manuscript is long, it took a substantial amount of time. I appreciate it.
Reviewer 4 Report
Comments and Suggestions for Authors
Second evaluation of manuscript nutrients-3633472
The authors' efforts to improve the quality of the text are noticeable. I will present my considerations regarding the new version of the text. In the previous version, I had asked that authors follow the instructions for authors from Nutrients. The improvement is noticeable, however, it is still not enough. To make the authors' work easier, Nutrients provides a template with the font format (Palatino linotype). I kindly ask the authors to format the text.
I would also like the authors to mark the changes made in some way (e.g. using a different font color), as this makes the reviewers' work easier. As it is, in order to locate the changes, I have to open the previous version of the text as well, which makes the reviewers' work more time-consuming.
The Figures have really improved a lot, but some of the Tables are still not in the journal format. I kindly ask the authors to review them.
I would also like to ask the authors to pay attention to the acronyms, such as LATAM and HC. They only appear twice in the text. Is it justified to create an acronym in this situation?
The authors establish a series of objectives at the end of the introduction, but what would be the benefit to the readers? How can this summary of content be useful to those who read this review in Nutrients? I kindly ask the authors to establish, at the end of the introduction, a justification that indicates the importance of this review being published.
In the first version, I asked the authors to better structure the review method. Even though it is a narrative review, it is important to state what the criteria were for an article to be included or not in the review. I see that this point was adequately addressed by the authors.
I believe that the rest of the text is well written, but the limitations should come before the conclusion.
Author Response
See file.
